# Behavioral Consequences Among Survivors of Cerebral Malaria and Acceptability to Different Disciplinary Methods

**DOI:** 10.3390/ijerph22060928

**Published:** 2025-06-12

**Authors:** Gudlaug Maria Sveinbjornsdottir, Sam Kabota, Sveinbjorn Gizurarson, Urdur Njardvik

**Affiliations:** 1Department of Psychology, School of Health Sciences, University of Iceland, Sæmundargata 2, 102 Reykjavík, Iceland; 2Pediatric Department, Nkhoma Mission Hospital, Christian Health Association in Malawi, Nkhoma P.O. Box 48, Malawi; samkabota@gmail.com; 3Department of Pharmaceutical Sciences, School of Health Sciences, University of Iceland, Hofsvallagata 53, 107 Reykjavik, Iceland; sveinbj@hi.is

**Keywords:** ADHD, cerebral malaria, children, Malawi, ODD, treatment acceptability

## Abstract

Cerebral malaria (CM) is a life-threatening disease that affects mainly children in sub-Saharan Africa. Studies have shown that children who survive CM are often left with neurological disabilities after recovery, such as behavioral changes similar to attention deficit hyperactivity disorder (ADHD) and oppositional defiant disorder (ODD). However, diagnosis and treatments for ADHD are limited in many places in Africa. The purpose of this study was to assess behavioral changes following CM infection in Children in Malawi and parents’ acceptability of behavioral treatments. Twenty-one parents of children who had survived CM were compared to forty parents from the general population. Assessment instruments included the Disruptive Behavior Rating Scale (DBRS), Treatment Evaluation Inventory-Short Form (TEI-SF), and ADHD symptom checklist. The results showed that the most acceptable treatments among parents in Malawi were interrupt/redirect, discussion, and medication. Parents of CM-surviving children were significantly more accepting of medication (*F*(1,59) = 7.92, *p* < 0.007). The majority of the children who survived CM were rated above the clinical cut-off for ADHD and ODD symptoms.

## 1. Introduction

In children, malaria can be a life-threatening disease. It is caused by the parasite Plasmodium falciparum and is mainly spread by mosquitoes. It was estimated that in 2022 there were about 249 million cases of malaria worldwide, which constituted 3.11% of the Earth’s population at that time. Also, from the same report, it was estimated that about 608,000 deaths worldwide were due to malaria, or a little more than one death per minute. About 462,000 children under the age of five died from malaria, or approximately one child every minute, which accounted for 76% of all deaths caused by malaria that year [1]. During the COVID-19 pandemic, children’s deaths increased by 69,000 due to disruptions in the provision of malaria prevention, diagnosis, and treatment [2].

Today, pregnant women receive prophylactic treatment, if possible, because if a woman becomes infected during pregnancy, her unborn child may also get infected [3,4]. An infection can also cause severe disease in the placenta [5]. In some cases, the parasite travels to the brain, where it can cause cerebral malaria (CM), which is one of the most severe, life-threatening, and complex stages of malaria. Unfortunately, children under the age of 5 years are most susceptible to CM [6]. The symptoms and sequelae are much more serious than uncomplicated malaria. A diagnosis of CM can include altered neurological functions, making it hard for children to recover. Unfortunately, even with treatment, CM can be fatal [7]. About 8.5–20% of children in Africa diagnosed with severe malaria die from CM [8,9].

CM can cause neurological deficits in children that recover from the disease. One of these deficits is behavioral changes similar to attention deficit hyperactivity disorder (ADHD) and/or oppositional defiant disorder (ODD), which seem to be the most common mental disorders following recovery from CM. About one of every four children who survive CM shows deficits, such as inattention and lack of working memory. These symptoms appear about six months up to two years after their recovery, leaving the child with persistent behavioral problems. However, the children who show these symptoms do not always receive psychiatric or psychosocial assessment or help [7,10,11,12,13].

There is limited knowledge on the pathophysiology of cerebral malaria as well as how it affects a child’s cognitive development and behavior. More research is needed in this area [7,14]. According to Birkbeck et al. (2010) the number of cases of disruptive behavioral disorders is unknown among Malawian children, and it seems that these disorders are not underdiagnosed in Malawi [15]. However, there seems to have been an increase in research on disorders in this geographical area in the last few years [16,17].

In 2016, Idro et al. carried out a study looking at the frequency of mental health disorders in children who survived CM and severe malarial anemia, compared to a control group who were not infected with malaria. They found that children who had CM were at a higher risk of developing mental health disorders after the sickness. About 10.4% of children who had survived CM were diagnosed with some kind of mental health disorders, compared to 4% of the children who became infected with severe malarial anemia, and 1.8% of the children in the control group [12].

Another study carried out by Idro et al. found that some of the consequences experienced after recovering from CM improved after a certain amount of time. At the same time, other deficits showed little improvement during the same period [7]. Some of the deficits seen shortly after the recovery appeared while the children were still in the hospital, while others emerged later after the children had recovered [7]. One of the deficits that seemed to improve following recovery from CM was visual impairment. It took the child 1–7 months to gain almost complete or normal vision. Other deficits seemed to have long-term sequelae, such as behavioral changes. Such problems were, e.g., ADHD, conduct disorder, and pervasive developmental disorder, or the child could be placed on the autistic spectrum [7]. The most common behavioral problem was ADHD behavior. These behavioral changes did not show up until some time had passed since recovery. These changes occurred within two weeks after recovering from CM at the earliest. Interestingly, some behavioral changes seemed to improve when they were treated with drugs such as haloperidol or methylphenidate [7].

These behavioral changes also impacted the children’s performance in school. They dropped out of school, and their teachers did not know how to respond to this new behavior. It also had a significant impact on family relations and communication as the stress levels within the families seemed to increase because of these behavioral changes [7].

When it comes to the diagnosis of ADHD, there is limited access to proper diagnosis in many places in Africa, probably resulting from a lack of knowledge of diagnostic tools or even a lack of knowledge of ADHD in general [18]. In 2020, the prevalence of African children and adolescents with ADHD was assessed in a meta-analysis which showed that many countries in Africa do not use proper diagnostic instruments when screening for ADHD [19]. Nevertheless, in 2022, a study was conducted in Malawi, where proper diagnostic instruments, such as the Kiddie Schedule for Affective Disorders and Schizophrenia (K-SADS), were used to evaluate mental health disorders, such as ADHD [17]. However, treatments or psychological support for those diagnosed with ADHD may also be limited in this region of the world, as can be seen in a review article by Kasu and Chikanza from 2022 [20].

Studies on treatment acceptability in Africa are also lacking, including in Malawi. Studies have shown differences between countries regarding the acceptability of approaches to children’s behavior problems. Studies have also shown that mothers whose children show behavior problems are more likely to accept treatment such as positive reinforcement, response costs, and medication [21,22,23]. A study that Dorsey et al. conducted in 2022, found good treatment acceptability among participants in Tanzania and Kenya where trauma focused cognitive behavioral therapy (TF-CBT) was used with children and adolescents [24]. In that study, the assessment of treatment acceptability focused on interactions between participants and research team as well as considerations of ethicality [24,25]. Other treatment acceptability measurements, such as Treatment Evaluation Inventory-Short Form (TEI-SF), focus on other aspects of treatment acceptability, such as the likability of a treatment strategy [25]. Studies focusing on parental acceptability seem to be lacking in Africa.

The purpose of this study was twofold. Firstly, it assessed treatment acceptability among parents in Malawi whose children had recovered from CM compared to parents in the general population. Secondly, it assessed if children who survived CM showed behavioral changes after their recovery. The children were visited about one year after recovery. This time was selected based on previous work by Birbeck et al. [15], who showed a time lag from recovery until the behavioral changes started occurring.

## 2. Materials and Methods

### 2.1. Ethical Approval

This study was approved by the Malawian National Health Science Research Committee (NHSRC), operated by the Ministry of Health of Malawi. The application was submitted in early October 2019 and approved on 7 February 2020 (Approval number: 2452). The study in Nkhoma was also approved by the management of CHAM (Christian Health Association in Malawi) who operated Nkhoma Mission Hospital.

### 2.2. Locations of Data Collection

The data collection was conducted in two different sites in Malawi. The first part took place in both residential and business areas in Lilongwe, the capital of Malawi. The data collection for the second part took place in Nkhoma, a small village about 70 km southeast of the capital.

### 2.3. Data Collection

In the first part of the study, forty parents who lived in Lilongwe were randomly selected and asked if they could participate in the research. This data collection was supposed to be a survey. However, due to illiteracy by many participants, it was conducted through interviews where no personal information was collected. Each participant had the opportunity to decline participation but all accepted. The only criterium for participation was being a parent. Demographic information on the parents who participated in Lilongwe is not available since no information on the participant’s ages or any information about their children was collected. All participants in Lilongwe happened to be mothers.

In the second part of the study, we identified all children at Nkhoma Mission Hospital in Nkhoma, Malawi who had recovered from CM one year prior to the data collection and fulfilled the following criteria (all five were used):The patient files confirm that the child was diagnosed with CM.Prior to admission, there was a history of fever, convulsions, coma, or loss of consciousness.Positive malaria rapid diagnostic test (MRDT) results are used to detect evidence of malaria parasites in the human blood.Lumbar puncture for cerebral spinal fluid, which is the only way to ensure that the sickness is caused by CM and not, for example, bacterial meningitis.Excluding other causes of convulsions, such as meningitis, hypoglycemia, or other causes.

This yielded a sample of 21 parents, both mothers and fathers. All the parents participating lived in small villages within a 10 km radius of the Nkhoma Mission Hospital. The children were aged from three to almost seven years, and the gender rate was 76% boys and 24% girls.

In both parts of the study, all the parents who were asked to participate accepted.

### 2.4. Questionnaires

For the data collection, three questionnaires were used. However, due to a high illiteracy rate in Malawi, the questions were asked in an interview format. Only one questionnaire, the TEI-SF, was used for the first part of the study, while all three questionnaires were used in an unstructured interview for the second part. A description of each of the questionnaires is listed below.

#### 2.4.1. ADHD Based on DSM-5

A questionnaire about ADHD symptoms was created for this study and used to evaluate behavioral changes that children might show after recovering from CM. The parents were asked how much time had passed since the diagnosis of the disease, whether they noticed any behavior changes after the illness, how long the changes lasted, and how they would describe the changes. These questions were chosen to see whether parents noticed any behavioral changes after recovery. Research has shown that behavioral changes may occur after a certain amount of time has passed since recovery and that these changes often last for a long period of time [7]. This questionnaire also contained a list of the ADHD symptoms from the *Diagnostic and Statistical Manual of Mental Disorders, 5th Edition* (DSM-5). These symptoms were used to evaluate behavioral issues in the children. The parents were asked if they noticed any behavioral changes similar to the ADHD symptoms after recovery, which they had not noticed prior to their sickness. The answers were categorized into two answers, “no, this behavior has not occurred after the recovery from CM” or “yes, this behavior has occurred after the recovery from CM.”

#### 2.4.2. Disruptive Behavior Rating Scale

To evaluate disruptive behavior, the Disruptive Behavior Rating Scale (DBRS) was used. It evaluates how frequently ODD symptoms have occurred within the last six months in children. The list has eight statements based on diagnostic criteria for ODD according to the *Diagnostic and Statistical Manual of Mental Disorders, 4th Edition* (DSM-IV). As a Chichewa version does not exist, we used the original English version of the DBRS and translated it to Chichewa during administration with the help of an interpreter. The internal consistency for the questionnaire has been shown to be good (α = 0.94) [26]. The Cronbach’s Alpha for the sample in this study was also good or α = 0.91.

#### 2.4.3. Treatment Evaluation Inventory

All the participants were asked questions from TEI-SF [27]. The questionnaire was used to assess treatment acceptability within the group of parents in the general population compared with parents whose children recovered from CM. As a Chichewa version does not exist, we used the original English version of the TEI-SF and translated it to Chichewa during administration with the help of an interpreter. The internal consistency for the TEI-SF has been shown to be very good (α = 0.85) [27].

Additionally, to the TEI-SF, the participants were told a short story about an 8-year-old boy who misbehaves, disobeys his parents, and irritates his little sister. After the participants heard the story, they were asked questions regarding seven reactions to the boy’s behavior. These reactions were: “spanking”, where the parents hit the child on its’ bottom with their palm; “time-out”, where the parents place the child in a corner of a neutral space for as many minutes as the age of the child; “response cost”, where the parents take away some privilege from the child when it disobeys; “differential attention”, where the parents either ignore the child if it disobeys or give the child praise and attention if it behaves in a good way; “medication”, which a doctor provides and are supposed to help the child to calm down and improve its’ attention; “discussion”, where the parents talk to the child about the behavior and the consequences that follow the behavior; “interrupt-redirect”, where the parents try to redirect the child with encouragement to finish tasks or with a choice to do another task instead. The parents were also asked if there would be another way that they would react to the boy’s behavior in the story.

The questions that followed the short story were about how willing the parents would be to use the above-mentioned reaction or methods, how effective they would be, and if they would be an acceptable way of treating a child’s behavior. There were nine questions regarding the acceptability of each method, which were answered on a Likert scale from 1 to 5, where 1 is strongly disagreed and 5 is strongly agreed.

### 2.5. Procedure

In the first part, a Malawian translator assisted with all communication, questions, and answers from English to Chichewa (the local language in Malawi), and vice versa, because of illiteracy and the lack of English knowledge among local participants. The translator was a student in physiotherapy. The parents were randomly selected from the streets of Lilongwe and asked by the translator if they were willing to participate in a research project by answering some questions about treatment acceptability.

In the second part, health professionals at Nkhoma Mission Hospital translated the communication, questions, and answers. They also assisted with the selection of the participants for the study. After a child had been identified, its parents were contacted by a hospital staff member and asked if they were willing to participate in this project. Everyone received the same opportunity to participate or refuse participation.

All participants signed informed consent after receiving information concerning the study. It had to be read for the parents, and many of them signed by using their thumbprint because of illiteracy. First, after signing, the questionnaire was presented.

### 2.6. Data Analysis

The software SPSS from IBM SPSS Inc. (version 26) was used to analyze the data. One-way ANOVA was used to analyze results from the TEI-SF, when comparing parents whose child had recovered from CM and parents in the general population. One-way ANOVA was also used to examine differences in the answers from parents of girls who had survived CM compared to parents of boys who had survived CM. The descriptive analysis included percentages, means, and standard deviation for all the questionnaires. When analyzing parental treatment acceptability in each group, the percentage was found using the TEI-SF by combining answers marked as 4 and 5 on the Likert scale. For DBRS, the answer options “often” and “very often” from the Likert scale were combined to be able to see how many parents were seeing the behavior often or very often.

## 3. Results

### 3.1. Treatment Acceptability

The outcome for the treatment acceptability is shown in Table 1 for each treatment method, using the TEI-SF. Each treatment’s mean score and standard deviation on the TEI-SF grouped by (a) parents of children that had recovered from CM and (b) parents in the general population is also shown in Table 1. The higher the score, the higher the acceptability was among the parents of the method. Discussion was rated most acceptable by both groups, while spanking was least acceptable among the general population, and time-out was least acceptable among the CM parents. The results showed that there was a significant difference between parents’ acceptability regarding using medication to treat misbehavior (*F*(1,59) = 7.92, *p* < 0.01), as parents whose child had recovered from CM found that method more acceptable. There was also a difference regarding time-out, although it was not statistically significant (*F*(1,59) = 3.5, *p* = 0.066). There were, however, no significant differences between the parents’ acceptability of the other treatment methods.

A comparison of acceptability for treatment methods, between parents whose boys had recovered from CM compared to parents whose girls had recovered from CM, showed no gender differences.

### 3.2. ADHD Symptoms

When the parents were asked about changes in their child’s behavior following CM, 81% stated their child had displayed six or more behavioral changes that comply with the criteria for the number of ADHD symptoms. The majority of the children showed more than six attention difficulties and/or hyperactivity/impulsivity symptoms. About half of the study group showed at least twelve symptoms, where at least six symptoms were associated with attention difficulties, and at least six were associated with hyperactivity/impulsivity. They reached, therefore, the clinical criteria for the number of symptoms for the combined subtype.

One child showed all symptoms after recovering from CM, according to their parent, and four children showed 17 out of 18 symptoms. Only 4 out of 21 children showed fewer than 6 symptoms, where 1 child showed no symptoms at all, and 1 showed only 1 inattention symptom, according to their parents.

Table 2 shows the percentage for each symptom in the study population, displaying the number of children showing each symptom and split by gender. Less than half the boys showed the following three ADHD symptoms: unable to play quietly (37.5%), talking excessively (43.8%), and difficulty sustaining attention (43.8%). At the same time, 80% of the girls showed each of these same symptoms. Six additional ADHD symptoms were expressed in 80% of the girls. All the girls showed the following five ADHD symptoms after recovering from CM: does not seem to listen, does not follow through on instructions, forgetful in daily activities, fidgets with or taps hand, and driven by a motor. Sixty percent of the girls showed the remaining four of the eighteen ADHD symptoms. None of the ADHD symptoms that the girls showed had an occurrence rate lower than 60%, which indicates that the majority of the girls showed ADHD symptoms after recovering from CM. However, it is important to note that there were only 5 girls compared to 16 boys.

In Table 3, the proportion of children who survived CM, and showed six or more ADHD symptoms after recovery, are shown. When divided by gender, girls showed a higher percentage based on six or more ADHD symptoms than the boys. All the girls and 12 out of 16 boys showed at least six symptoms of ADHD. All of those 12 boys showed at least 6 hyperactivity/impulsivity symptoms after the recovery, where the same applied to 4 of the 5 girls. Also, 4 of the 5 girls showed at least 6 attention difficulties symptoms, while only 8 boys showed them. In total, 50% percent of the boys and 60% of the girls showed that cerebral malaria resulted in the number of symptoms needed for the combined subtype of ADHD.

### 3.3. ODD Symptoms

In this study, 14 children out of 21 (3 girls and 10 boys) had a score above the clinical cut-off score for ODD symptoms based on the DBRS. That accounts for 67% of the entire study group. This can be seen in Table 3, as well as how many of the children who survived CM showed the minimum clinical criteria for both ODD symptoms and ADHD symptoms after the recovery, divided by subtypes.

In most cases, a higher percentage of boys met the clinical criteria in a number of symptoms for ODD and ADHD compared to the girls, who were fewer in number, which makes it difficult to evaluate gender differences. However, when evaluating combined ODD with only attention difficulties, a higher percentage of the girls than boys met the clinical criteria for these symptoms. Most children that reached the criteria for ADHD symptoms also met the criteria for ODD symptoms.

All boys, except one, who showed at least six symptoms for any subtype of ADHD also reached the clinical criteria for ODD after the recovery. However, three out of five girls showed at least six ADHD symptoms in addition to the clinical criteria for the number of ODD symptoms.

Table 4 shows the proportion of children showing the ODD type of behavior about six months prior to the data collection. The table shows the results for all children and for each gender. The majority of children showed many of the ODD symptoms, often or very often. Sixty percent of the girls showed each symptom, except for “spiteful or vindictive”, where only 40% showed that symptom. Most of the boys showed “is easily annoyed” (81.3%) and “deliberately annoys” (75.1%).

## 4. Discussion

### 4.1. Treatment Acceptability

No prior studies on treatment acceptability regarding parents’ management of behavioral problems in Malawi were found. In fact, studies on this subject seem to be lacking in African countries, although there has been some research in recent years. It seemed that almost all participants, both parents of children who had survived CM and parents in the general population, thought of discussion as an acceptable treatment when disciplining children. Interrupting and redirecting were also viewed as acceptable methods. Both groups viewed spanking as an unacceptable way to discipline children.

There were some differences when the results were studied for each group (for parents in the general population exclusively and parents whose children recovered from CM exclusively). Interestingly, the acceptability towards the use of medication was significantly different between these groups, where 95% of the CM parents viewed it as an acceptable method, but only 65% of parents in the general population. This difference might reflect that CM parents that almost lost their child due to CM experienced first-hand how pharmacological treatment could save lives and help their children. Therefore, they might put more trust in medicines than the general population. The parents in the general population were randomly selected, and some of them might never have experienced pharmaceutical interventions. Earlier studies showed that mothers whose children show severe behavioral problems are more likely to accept methods such as medication [22]. It might also be the case in this study that the severity of the child’s behavior problems led the parents, whose children had survived cerebral malaria, to view medication as a more acceptable way of treating children.

The intervention time-out was not rated as an acceptable method among CM parents (1.6%), where about 14.7% of the general population found this method acceptable. Although the average score was different, it did not reach a significant difference, possibly due to the small sample size and small statistical power.

Many CM parents viewed differential attention as both a good and bad method. They viewed the method as good since it gave the child praise and because attention stimulates good behavior. However, they did not like the method because they said it is not good to ignore the child when he or she disobeys them. Earlier studies have shown that behavioral treatments have yielded good and effective results when used on children with ADHD. One such treatment includes giving praise and attention to them and ignoring children when they show problematic behavior [28]. This method, however, did not gain much acceptance among the parents in Malawi.

This study is in line with the results from earlier studies, where parents found the use of positive reinforcement, medication, and response cost as acceptable methods, rather than methods such as spanking and time-out [21,22]. It is, therefore, interesting that only a small fraction of the parents in the general population of Malawi accepted methods like spanking (7.5%) and time-out (14.7%). There was also no difference between parents who had boys or girls, suggesting that parents do not base their intervention on gender in the group of parents whose child survived CM, which might be because of imbalance in number of genders.

When conducting the study, many parents in both groups discussed other methods they would prefer to use when responding to their child’s misbehavior. For example: sending the child to school so they will learn the appropriate way to behave; sending the child to church to learn how to behave; praying for the child; and consulting the child. It was more common among parents in the general population to suggest going to church or praying for their child. At the same time, it was more common among the parents whose child had recovered from CM to want to discuss, teach and counsel the child on how to behave.

Interestingly, when comparing the results from this study with those of a study carried out by Njardvik and Kelley in 2008, the parents in both groups (in Malawi) thought that it would be more acceptable to use medication, interrupt/redirect, and discussion than parents in Iceland and the United States. However, the parents in Malawi thought it was less acceptable to use time-out and differential attention compared to Icelandic parents and parents in the United States. Parents in both Malawi and Iceland were less accepting of the use of spanking, response cost, and punishments in general compared to parents in the United States [23]. These results indicate interesting cultural differences between these countries regarding treatment acceptability. The results support the idea that cultural differences can be an essential factor when it comes to treatment acceptability for behavioral problems.

### 4.2. ADHD Symptoms

The results show that CM can have long-term consequences for the children who survive the disease. The pathophysiology of the disease inside the brain is not fully understood [14]. However, this study indicates that children who have recovered from CM show a clear behavioral change. According to their parents, the majority of children showed six or more ADHD symptoms, which is the clinical criterion for the number of symptoms.

When reviewing individual ADHD symptoms, the results showed that more than half of the study population expressed the majority of symptoms that are used to diagnose ADHD. This indicates that children seem to be more likely than not to show ADHD symptoms after recovering from cerebral malaria. Both attention difficulties and hyperactivity/impulsivity symptoms were found to be more common after the recovery, and about half of the sample population reached clinical criteria for the number of symptoms required for the combined subtype of ADHD.

The percentage of children who showed six or more ADHD symptoms after recovering is higher than expressed in other studies. Birbeck et al., from 2010, showed that only ten percent of the children started showing ADHD symptoms following recovery from CM [15]. Similarly, in a study by Idro et al., from 2016, about 10% of the children who survived showed some mental health disorders after recovery [12]. The difference might be due to the fact that, in this study, the parents were only asked if they had noticed symptoms occurring after the recovery, but not how frequently the symptoms occurred. Therefore, it is unclear whether the symptoms occurred occasionally or how inhibitory they are. Thus, while the children showed many symptoms, the symptoms may not have been severe enough to cause impairment in their daily lives.

The results indicated that there was a greater variance in the percentage of ADHD symptoms among boys than among girls, but this should be interpreted with caution as there were much fewer girls than boys in the sample. The proportions of girls that showed each ADHD symptom was, in general, higher than in boys. All the girls reached clinical criteria based on the number of ADHD symptoms. There seemed to be more diffusiveness regarding neurological impairment among boys. Attention difficulties seemed to be more common among girls than boys. That might be because there were fewer girls than boys in the study population. The results show that the majority of children, irrespective of gender, seem to show ADHD symptoms after recovery from CM. This is not in line with previous studies where for example, Birbeck et al. found that boys were at higher risk of developing neurodisabilities than girls [15].

### 4.3. ODD Symptoms

A similar outcome was shown when looking at ODD symptoms in children that survived CM. Many of the children also reached the clinical criteria for ODD about six months after the recovery. Furthermore, a high percentage of children reached the clinical criteria for both ODD symptoms and ADHD symptoms. Over half of the children showed both ODD and six or more symptoms of attention difficulties and/or hyperactivity/impulsiveness. However, little less than half of the children reached the clinical criteria for both ODD and the combined subtype of ADHD. This was the case for both genders. Only one symptom, loses temper, was below 50% in occurrence. The majority of children who showed the symptoms “is easily annoyed” and “deliberately annoys” others.

These results align with previous studies where it has been shown that children who recover from CM sometimes show ODD symptoms after their recovery. Idro et al. found that the most common mental disorders that occurred following CM are externalizing disorders like ODD and ADHD [12].

### 4.4. Limitations and Further Studies

One of the limitations to this study was that all the questions and answers were translated from English to the local language, Chichewa, and then back to English via a translator, and thus we were unable to follow the guidelines for translating questionnaires from the International Test Commission (International Test Commission, 2017) [29]. Also, the parents’ and the translators’ understanding of the questions was sometimes limited, which may have affected the translation. Social desirability might also have influenced some of the answers, especially by parents whose child recovered from CM. Here, the translators were staff members from the local hospital, where their child was admitted. There is a possibility that the parents might have wanted to give answers that they assumed were expected. Another weakness of the study is that the sample size was very small, and no demographic or socioeconomic data was gathered from the participants. Hence the representativeness of the sample is unknown and the generalization of the results is limited. Nonetheless, the results of the study indicate that behavioral difficulties are common among children surviving CM and further research in this area seems warranted.

Future studies are needed on CM to understand these changes. Both in pathophysiological studies, to understand where the parasite goes inside the brain, and how it causes these changes, as well as to understand the psychological and neurological changes that seem to appear following the recovery. As our sample size was small, it would be interesting to repeat this study in a much larger sample targeting larger geographical areas. Additionally, it will be important to follow up on this work by providing training to parents, health professionals, and children on how to train and discipline children immediately after they have recovered in order to evaluate if this behavior can be prevented or changed. It would also be interesting to see whether parents’ mental health or trauma that they might have experienced may have influenced how they view their children and their behavior.

## 5. Conclusions

Hopefully, this study will provide an important steppingstone towards understanding the sequalae of cerebral malaria and how the parasite induces behavioral changes in children who survive this life-threatening disease. This work may also provide a basis for suggesting and developing appropriate teaching and training for parents and health care workers in the sub-Saharan region of Africa to help them understand and work on behavioral changes that they may experience when their child comes back home. This may include clinical guidelines and/or psychological treatment for this group of children and their parents.

One of the comments received was that there is a tendency to criticize these parents for not disciplining their children, or in some cases families believe that the disease has resulted in an evil spirit occupying their child. Such issues need to be discussed in a way that these people understand how the disease may change the behavior of children over the coming months. These parents need support rather than critics.

## Figures and Tables

**Table 1 ijerph-22-00928-t001:** Proportion of parents that accept each disciplinary method, and mean ratings ± SD in parentheses, for the Treatment Evaluation Inventory-Short Form (TEI-SF), divided by parents in the general population and parents whose child recovered from cerebral malaria (CM).

Treatment	General Population (*n* = 40)		CM Parents (*n* = 21)		Statistical Analysis and *p*-Value	
	Proportion (%)	Mean Ratings ± SD	Proportion (%)	Mean Ratings ± SD		
Spanking	7.5%	11.8 ± 9.2	2.7%	10.2 ± 1.8	*F*(1,59) = 0.61	*p* = 0.44
Time-out	14.7%	14.3 ± 11.5	1.6%	9.7 ± 1.7	*F*(1,59) = 3.5	*p* = 0.067
Response cost	48.9%	26.6 ± 17.5	41.3%	25.7 ± 16.1	*F*(1,59) = 0.31	*p* = 0.59
Differential attention	12.5%	13.2 ± 11.8	23.4%	17.9 ± 14.3	*F*(1,59) = 1.51	*p* = 0.24
Medication	65.3%	32.5 ± 17.0	95.2%	43.3 ± 6.2	*F*(1,59) = 7.92	*p* < 0.007 **
Discussion	99.4%	44.7 ± 1.4	97.9%	46.3 ± 10.6	*F*(1,59) = 1.99	*p* = 0.17
Interrupt/redirect	91.9%	42.1 ± 9.6	90.0%	41.6 ± 10.3	*F*(1,59) = 0.062	*p* = 0.80

** *p* < 0.01. Note: The higher the score, the more positive the rating is.

**Table 2 ijerph-22-00928-t002:** Proportion of children showing behavior related to attention deficit hyperactivity disorder (ADHD) about 1 year after recovering from cerebral malaria (CM).

Questions	All (%)	Boys (%)	Girls (%)
	(*n* = 21)	(*n* = 16)	(*n* = 5)
Attention difficulties			
Fails to give close attention	61.9	56.3	80.0
Difficulty sustaining attention	52.4	43.8	80.0
Does not seem to listen	61.9	50.0	100.0
Does not follow through on instructions	66.7	56.3	100.0
Difficulty organizing tasks	57.1	50.0	80.0
Avoids tasks that require much mental effort	71.4	68.8	80.0
Loses things	66.7	68.8	60.0
Easily distracted	66.7	68.8	60.0
Forgetful in daily activities	66.7	56.3	100.0
Hyperactivity/impulsivity			
Fidgets with or taps hands	81.0	75.0	100.0
Leaves seat	71.4	68.8	80.0
Runs or climbs	71.4	75.0	60.0
Unable to play quietly	47.6	37.5	80.0
“Driven by a motor”	76.2	68.8	100.0
Talks excessively	52.4	43.8	80.0
Blurts out answers	61.9	62.5	60.0
Difficulty waiting	76.2	75.0	80.0
Interrupts or intrudes	76.2	75.0	80.0

**Table 3 ijerph-22-00928-t003:** Proportion (%) and numbers (*n*) of children who survived cerebral malaria (CM) and fulfil the clinical criteria for attention deficit hyperactivity disorder (ADHD) and oppositional defiant disorder (ODD) symptoms.

Symptoms	All (21)	Boys (16)	Girls (5)
Attention deficit hyperactivity disorder (ADHD)	81% (17)	75% (12)	100% (5)
Attention difficulties	56% (12)	50% (8)	80% (4)
Hyperactivity/impulsivity	76% (16)	75% (12)	80% (4)
Combined subtype	52% (11)	50% (8)	60% (3)
Oppositional defiant disorder (ODD)	67% (14)	69% (11)	60% (3)
ODD + attention difficulties	52% (11)	50% (8)	60% (3)
ODD + hyperactivity/impulsivity	62% (13)	69% (11)	40% (2)
ODD + combined subtype	48% (10)	50% (8)	40% (2)

**Table 4 ijerph-22-00928-t004:** The proportion of children who survived cerebral malaria that showed oppositional defiant disorder (ODD) symptoms about six months after recovery from their disease.

**Behavior**	**All (%)**	**Boys (%)**	**Girls (%)**
	**(*n* = 21)**	**(*n* = 16)**	**(*n* = 5)**
Loses temper	47.7	43.8	60.0
Argues with adults	61.9	62.6	60.0
Refuses to comply	61.9	62.6	60.0
Deliberately annoys	71.4	75.1	60.0
Blames others	61.9	62.8	60.0
Is easily annoyed	76.2	81.3	60.0
Angry or resentful	61.9	62.6	60.0
Spiteful or vindictive	52.4	56.3	40.0

## Data Availability

The data presented in this study are available on request from the corresponding author, G.M.S.

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
