# Peer review of "Behavioral Consequences Among Survivors of Cerebral Malaria and Acceptability to Different Disciplinary Methods"

_ijerph, 2025, doi:10.3390/ijerph22060928_

Round 1
Reviewer 1 Report
Comments and Suggestions for Authors
Thank you for the opportunity to review "Behavioral Consequences Among Survivors of Cerebral Malaria and Acceptability to Different Disciplinary Methods."
This manuscript presents information regarding survivors of cerebral malaria and the subsequent behavioral symptoms associated with the condition. These authors also present parents' perceptions of different treatment options that may be applicable to their child.
An area for improvement in this paper is clarity regarding what the manuscript is presenting. The title leads the reader to believe there is going to be treatment provided to the population, but no treatment was provided. It's more of an exploratory analysis of parents perceptions of their child's behaviors.
Additionally, given the exploratory nature of this information, a more robust future directions section would be appropriate. For example, what are the clinical implications of this information and what should be done as a follow up to support these families?
While I think the authors are bringing to light an important topic, I think the overall manuscript needs more development and clarity.
Reviewer 2 Report
Comments and Suggestions for Authors
Dear authors,
Thank you for your submission.
This is an important work! I have enjoyed reading your paper. Overall, it is well-written and informative. The research study is well-planned and well-implemented.
I have some review comments, recommendations, and suggestions to improve overall readability.
Abstract
DBRS; TEI-SF: please define what these abbreviations stand for as this is the first time they appear in text.
"Parents of CM surviving children were significantly more accepting of medication." Please add significance level (p value)
Keywords: it is recommended to arrange keywords in an alphabetical order.
Materials and Methods
Data collection: "a sample size of 21 parents participated" please report how did you determine sample size? were any sample size calculations performed?
Questionnaires: were those questionnaires pilot-tested and validated? please elaborate.
Procedure: according to evidence-based practice, it is recommended that the translation be applied in several stages by adopting the method of knowledge extraction, forward-backward translation, and experimental translation by specialized committees composed of translators specialized in both the language and topic, to ensure that the translated content is culturally suitable on one hand, while preserving scientific meaning on the other hand. I recommend adding this piece of information for better clarification and readability and provide more information about the translators in the current study who did the translation from English to Chichewa (the local language in Malawi) (i.e., their scientific background, level of expertise, how many health professionals did the translation during the second part of the study, etc.).
Discussion
Limitations and further studies: how about the relatively sample size used in the current study? was it representable enough for the target population and how did that affect generalizability of results in the current study? any recommendations for future studies? please elaborate.
Also, in the current study the gender rate was 76% boys and 24% girls. How do you think this affected the results? relatively equal gender distribution maybe preferred for better representation of the targeted population? please explain.
Abbreviations: please add DSM-5 abbreviation
Many thanks.
Best wishes,
Reviewer 3 Report
Comments and Suggestions for Authors
This is a cross-sectional study of parents whose children had cerebral malaria one year earlier to investigate incidence of ADHD and ODD symptoms as well as views on appropriate management in comparison with the general population. The background is well written and justifies the study, the method and results are clearly presented and the findings are sensitively discussed. I hope the suggestions below may help to strengthen this useful paper.
INTRODUCTION
The Introduction gives a very clear review of the literature in the area and well justifies the need for this study. No suggestions for changes here.
METHODS
- A query about eligibility criteria in part 1. It says, “The parents were randomly selected from the streets of Lilongwe and asked by the translator if they were willing to participate in a research project by answering some questions about treatment acceptability.” Were there any eligibility criteria? Did the interviewer and translator choose women only? (It says “parents” but that they were all mothers.) People within a certain age range? Did they ask if they had children, and exclude any who didn’t?
- “Only one questionnaire was used for the first part, while all three questionnaires were used in an unstructured interview for the second part.” Reading on to the Procedure section, as I understand it, the “one questionnaire” in the first part was the TEI-SF. Please give the name of that one questionnaire in the statement I’ve quoted.
- The descriptions of the contents of the questionnaires are very clear, but could the questionnaires themselves be included in supplementary materials, so that the reader can see exactly what questions were asked?
- Data analysis: With only 2 groups to compare, I would expect an independent samples t-test rather than a one-way ANOVA. The p values, however, will come out the same.
RESULTS
- Do you have any data on response rates? How many were invited (in each group) and what percentage of those participated? Did any begin and then withdraw?
- Remove this sentence: “This section may be divided by subheadings. It should provide a concise and precise description of the experimental results, their interpretation, as well as the experimental conclusions that can be drawn.”
- The results talk about “changes in their child’s behavior following CM”—but the results give the results only at 1 year post CM, so it is difficult to assess what the children’s symptoms were earlier than that. So, if I’m understanding this correctly, the study doesn’t seem to allow any statement about “changes”.
- Why are there 3 times more boys than girls in the CM group?
DISCUSSION
- The following statements suggest that you may have collected some useful qualitative data. Do you think that quotes from the participants might enrich the results? “They viewed the method as good since it gave the child praise and because attention stimulates good behavior. However, they did not like the method because they said it is not good to ignore the child when he or she disobeys them.” “Many parents in both groups discussed other methods 386 they would prefer to use when responding to their child´s misbehavior.”
- Change “sending the child to school, so it will learn the appropriate way” either to “sending the child to school, so he or she will learn the appropriate way” or “sending the child to school, so they will learn the appropriate way” to avoid using the pronoun “it” for the child.
- In the paragraph beginning, “Comparing symptoms by gender,” it would be wise to be cautious in interpreting these differences. As you point out in the Results section, there were only 5 girls.
- One limitation of this study that should be mentioned is the question of equivalence between the groups. No demographic or socioeconomic data are given, so it is impossible to know how similar the groups were. Having said that, the responses were so similar between groups, that this limitation seems less important than it would be, if you were basing conclusions on group differences. But the limitation should still be mentioned.
